



**Drivers of the δ¹⁸O Changes in Indian Summer Monsoon Precipitation between**
**the Last Glacial Maximum and Pre-industrial Period**
Thejna Tharammal [1*], Govindasamy Bala[2,] Jesse Nusbaumer[3]
[1] Interdisciplinary Centre for Water Research, Indian Institute of Science, Bengaluru
560012, India
[2] Centre for Atmospheric and Oceanic Sciences, Indian Institute of Science,
Bengaluru 560012, India
[3] National Center for Atmospheric Research, Boulder, USA
Corresponding author: thejnat@iisc.ac.in





**Abstract**
In this study, we investigate the changes in water isotope ratios in the Indian summer
monsoon precipitation ($\delta^{18}O_{precip}$) during the Last Glacial Maximum (LGM, ~21 ka
Before Present) compared to the pre-industrial (PI) period, and the mechanisms
driving these changes, using a general circulation model with water isotope and
novel water vapor source-tagging capabilities.
During the LGM, the model simulates a substantial reduction (15%) in monsoon
precipitation over the Indian subcontinent, consistent with proxy records. This drying
in LGM is associated with reduced atmospheric water vapor, a thermodynamic
response to cooling, while the westerly circulation, a dynamics response, is
strengthened over parts of the subcontinent. Additionally, zonal temperature
gradients between a relatively less-cooled tropical Western Pacific Ocean and Indian
subcontinent lead to anomalous subsidence over the Indian region, enhancing the
drying. Water vapor source tagging shows that while the four dominant moisture
sources for the monsoon (South Indian Ocean, Arabian Sea, Indian land recycling,
and Central Indian Ocean) remained the same, their contributions were reduced
during the LGM. The $\delta^{18}O_{precip}$ values over the Indian monsoon region are enriched
by approximately 1‰ in the LGM simulation, and we find that this enrichment was
not driven by the local amount effect. A decomposition analysis shows that the
enrichment was primarily caused by reduced contributions from distant, isotopically
depleted water vapor sources and secondarily by reduced rainout during moisture
transport from the Indian Ocean.
These findings have important implications for paleoclimate reconstructions,
suggesting that $\delta^{18}O$ records from the Indian region could be indicators of broad-
scale atmospheric circulation rather than being direct proxies for local precipitation
amount.






## 1. Introduction

The Indian summer monsoon (ISM) system, occurring during the months of June-September, is one of the major climate features in the world. It sustains the livelihoods of more than a billion people in the subcontinent by contributing almost 80% of the annual precipitation in the region. Monsoons were historically viewed as regional sea breezes driven by differential heating of land and sea. However, they are now understood as interconnected components of a global monsoon system, driven by the migration of Intertropical Convergence Zone (ITCZ), influencing tropical and subtropical precipitation (Gadgil 2003; Geen et al. 2020).

Recent decades have seen increasing ISM intensification, characterized by more frequent extreme rainfall events and increased variability, likely due to anthropogenic climate change (Krishnan et al. 2016, 2020; Wang et al. 2020; Chen et al. 2020; Katzenberger et al. 2021; Kong et al. 2022; Mukherjee et al. 2024). However, there is considerable uncertainty in these future projections (Krishnan et al. 2020). Even small changes in the monsoon patterns can adversely affect the annual rainfall (~10% reduction in the ISM precipitation from the mean is classified as drought; Shewale and Kumar 2005), hence, it is important to understand the changes and variability in the monsoon rainfall. Paleoclimate studies using climate archives and proxy records provide crucial constraints for reducing uncertainties in future climate projections (Tierney et al. 2020; Lohmann et al. 2020; Rehfeld et al. 2020; Brovkin et al. 2021; Kageyama et al. 2024). They are useful to understand the sensitivity of the monsoon systems to climate factors such as changes in greenhouse gases (GHG), orbital parameters, continental ice sheets, and Sea Surface Temperature (SST).

The Last Glacial Maximum-LGM, about 23000 to 19000 years before present, is a period of high interest in climate change studies. LGM presents a valuable case study for understanding how the Earth's climate system responded to the influence of reduced $CO_2$, presence of the Laurentide ice sheets and ice-sheet topography, and orbital forcing. The abundance of proxy-climate records for this period facilitates comparisons between proxy data and models. Climate records from the Indian Summer Monsoon region such as, water isotope records from sedimentary cores and speleothems (Contreras-Rosales et al. 2014; Sinha et al. 2015; Kathayat et al.



2016; Liu et al. 2021) indicate that LGM was characterised by a weaker Indian
Summer Monsoon. Climate modeling studies suggest that the general reduction in
precipitation over the globe during the LGM is mainly due to the cooling driven by
lower greenhouse gas concentrations and the expansion of ice sheets (Broccoli and
Manabe 1987, 2008; Yanase and Abe-Ouchi 2007; Tharammal et al. 2013;
Kageyama et al. 2020; McGee 2020; Seltzer et al. 2024). Further, the associated
cooler sea surface temperatures in the tropics during the LGM (MARGO Project
Members 2009; DiNezio et al. 2018; Tierney et al. 2020) likely influenced the
strength of the monsoon circulation, weakening the precipitation.

Stable isotopes of water undergo temperature-dependent fractionation during phase
changes. The resulting variations in the ratios of heavy to light isotopes (δ-values)
serve as powerful tracers of hydrological and atmospheric processes (Galewsky et
al. 2016; Dee et al. 2023; Bailey et al. 2024). Records of water isotopes in the
climate archives such as speleothems, tree rings, and sediment records are one of
the major proxies in reconstructing the Indian monsoon precipitation (Yadava et al.
2004; Maher 2008; Contreras-Rosales et al. 2014; Sinha et al. 2015; Kathayat et al.
2016). To interpret these climate records, *the amount effect* (Dansgaard 1964),
which is the observed inverse relationship between the ratio of water isotopes in
precipitation ($\delta^{18}O_{precip}$) and the amount of precipitation in the tropical regions, is
used. The amount effect is related to depletion of water vapor of heavier isotopes
during intense precipitation events, especially in convectively active tropical
monsoon regions (Risi et al. 2008; Lee et al. 2008; Tharammal et al. 2017).
However, the local precipitation amount is not the only factor that determines the
isotopic composition of precipitation in the tropics. It is also influenced by other
factors such as, relative contributions of moisture from various water vapor sources,
atmospheric circulation, and upstream convection (Lewis et al. 2010; Breitenbach et
al. 2010; Pausata et al. 2011; Sjolte and Hoffmann 2014; Zhu et al. 2017; Tabor et
al. 2018; Konecky et al. 2019; Hu et al. 2019; Tharammal et al. 2023; Chakraborty et
al. 2025). This complexity leads to considerable uncertainty in interpreting $\delta^{18}O$
records from the ISM region. Further, uncertainties remain in inferring the water
isotope proxy records due to low data resolution and sparse coverage.



While proxy records have been used to study past changes in the Indian monsoon
(Yadava et al. 2004; Dutt et al. 2015; Sinha et al. 2015; Kathayat et al. 2016;
Kaushal et al. 2018), climate model studies focussed on the Indian monsoon
precipitation and water isotope ratios during the LGM are largely lacking. Recent
advancements in climate models equipped with water isotope tracers in their
hydrology, along with the capabilities of tracking the evaporative water (Brady et al.
2019) will enable us to find the climatic factors affecting the $\delta^{18}O_{precip}$ of monsoon
precipitation, and also differentiate the moisture sources and their effects on the
$\delta^{18}O_{precip}$ (Hu et al. 2019; Kathayat et al. 2021; He et al. 2021; Tharammal et al.
2023). Therefore, applying these novel modeling techniques to resolve the drivers of
$\delta^{18}O_{precip}$ change in the Indian region during the LGM is a key research gap that this
study aims to fill.

In this study, we examine the mechanisms behind the changes in the monsoon
precipitation and water isotope ratios in the ISM region during the LGM using a
climate model with water isotope and novel water vapor source-tagging capabilities.
We will analyse the responses of water isotopes in precipitation, and moisture
sources to the glacial climate, and importantly, identify the major physical processes
influencing the changes in the isotopic ratios of precipitation. We will analyse the
relative importance of the "amount effect" compared to changes in moisture source
and transport in influencing the $\delta^{18}O_{precip}$ values during the LGM.

The paper is structured as follows: In Section 2, we introduce the model simulations
and methods. Section 3 includes the assessment of model performance under the PI
climate, analysis of changes in the monsoon, water vapor sources, and isotope
ratios of precipitation in the LGM simulation. Section 4 includes a discussion of the
results and main conclusions of the study.

**2. Methods**
**2.1 Climate Model**
Our study uses the Community Earth System Model, CESM version 1.2 with water
isotope tracking capabilities (iCESM, Brady et al. 2019) from the National Center for



Atmospheric Research (NCAR) for the climate simulations. Atmospheric and land
components of the iCESM are isotopic versions of the Community Atmosphere
Model CAM version 5.3 and Community Land Model CLM version 4, respectively.
The sea-ice model in the iCESM is Los Alamos Sea Ice Model version 4 (CICE4),
which is run in prescribed sea ice mode for the simulations presented here. The
isotope tracking in the model is facilitated with the inclusion of a parallel hydrologic
cycle for the water isotope tracers in the iCESM. It follows the water isotope ratios,
fluxes, and isotopic fractionations on phase changes in the components of the
hydrologic cycle (Brady et al. 2019).
iCESM has proven successful in reproducing the present global distribution of
isotopes in precipitation (Brady et al. 2019). Further, the model includes a tagging
feature for the evaporated water and can be used to track the sources of water vapor
for precipitation in a sink region. The model was successfully used in several studies
to reconstruct the past and present climate and isotope ratios in precipitation, and to
track the sources of water vapor in various tropical regions (Tabor et al. 2018; Hu et
al. 2019; Windler et al. 2020; He et al. 2021; Tharammal et al. 2023).

## 177    2.2 Experiments

We conducted two time-slice simulations for the current study, a) the pre-industrial
(PI) control experiment and b) the LGM simulation, with prescribed SSTs, sea ice
extent, and prescribed ocean surface isotopic ratios.
The isotopic composition of meteoric water is represented by the delta (δ) value in
permil (‰) units in the paper, denoting the relative abundance of the ratio of heavy
isotope to the light isotope in a sample with respect to a geochemical standard
(VSMOW-Vienna Standard Mean Ocean Water).
Accordingly, $\delta^{18}O$ is $(R_{sample}/R_{VSMOW} - 1) \times 1000$. R is the ratio of heavy to the light
isotope, $^{18}O/^{16}O$. $R_{VSMOW}$ is the standard isotope ratio.

For the PI simulation, the orbital conditions, GHG, SST, sea ice extent, and aerosol
boundary conditions are set at the year 1850. The GHG and orbital boundary
conditions of the experiments are given in Table S1. The SST and sea ice fraction
data for the PI experiment are derived from the corresponding coupled CESM





simulation (Zhu and Poulsen 2021). A uniform sea-surface δ¹⁸O of 0.5‰ is
prescribed for the control simulation. This is an approximate value based on present-
day observations and is close to the observed surface values of the tropical and
subtropical oceans (Hoffmann and Heimann 1997; LeGrande and Schmidt 2006; Lee
and Fung 2008).
For the LGM simulation, we follow the Paleoclimate Modelling Intercomparison
Protocol version 4 (PMIP4; Kageyama et al. 2020), and the GHG, orbital parameters,
land-sea mask, and surface topography are set to 21 ka BP conditions. The ice
sheet extent and topography for the LGM experiment (Fig. S1a) are derived from the
ICE-6G ice sheet reconstructions by Peltier et al. (2015). The coastlines for the LGM
experiment are adapted from the ICE-6G reconstruction and represent a lowering of
sea level by 120m during the LGM (Lambeck et al. 2014). The SST and sea ice
fraction for the LGM simulation (Fig. S1b, c) are obtained from the CESM coupled
LGM simulation (Zhu and Poulsen 2021). The formation of large continental ice
sheets during the LGM led to enrichment of heavier isotopes in the seawater oxygen
isotope ratios (ice-volume effect, Lambeck 2000). It is widely accepted that the sea
surface water isotope ratios during the LGM were approximately 1‰ enriched
compared to the pre-industrial values (Sima et al. 2006; Duplessy et al. 2002). We
represent this in the LGM simulation by prescribing a uniform sea surface
enrichment of water isotopes by 1‰ for δ¹⁸O, compared to the PI value.
Further, to identify the effects of water vapor sources on the monsoon precipitation
and water isotopes during the LGM, we tag the evaporated vapor from 17 ocean and
land regions in and around the ISM region in both the PI and the LGM experiments
(tagged regions are shown in Fig. S2a). The simulations are run for 30 years, and
the last 20 years are used for the analysis.

**2.3 Monsoon circulation indices and moisture budget analysis**

*2.3.1 Monsoon circulation indices*

Strength of the monsoon circulation can be estimated using various indices (Li et al.
2024). We calculate the monsoon circulation strength in PI and LGM simulations
using the following six indices selected to capture both circulation changes and water
vapor transport related to the monsoon precipitation. The geographical domains
used for the estimation of these indices are shown in Figure S2b.



1) The Somali jet index (Boos and Emanuel 2009), which is calculated as the square
root of twice the domain mean kinetic energy ($\sqrt{2\overline{KE}}$) of the 850 hPa horizontal
wind over the region, [5°S - 20°N, 50°E - 70°E].

2) The hydrological index, following (Fasullo and Webster 2003), calculated by
averaging the Vertically Integrated Moisture Transport (VIMT) in the Indian Ocean-
Arabian Sea region, [20°S-30°N, 40°E-100°E]. VIMT is the total horizontal movement
of water vapor in a vertical column of the atmosphere, and we calculate the term
from the surface up to 300 hPa.

$$VIMT = \frac{1}{g} \int_{P_{surface}}^{P_{top}} qV\,dp$$

234 ----------------- (1)


Magnitude of VIMT is,

$$|V\vec{IMT}| = \sqrt{VIMT_u^2 + VIMT_v^2}$$

237 ----------------- (2)

Where P is atmospheric pressure, g is gravity, q is the specific humidity, and V is the
wind vector with zonal and meridional components u and v.
3) Mid-tropospheric temperature gradient (ΔTT), defined as the tropospheric
temperature difference between a northern region and a southern region in the larger
monsoon domain (Xavier et al. 2007). ΔTT signifies the cross-equatorial temperature
gradient, and the onset of the ISM is defined as the time when ΔTT changes from
negative to positive.
4) The vertical shear of zonal winds, following (Webster and Yang 1992), calculated
as the difference between 200 hPa and 850 hPa zonal winds (U200-U850),
averaged over the region 10°N-30°N, 50°E-95°E.
5) The meridional shear of the 850 hPa zonal wind (barotropic shear, -∂u/∂y) over the
region 10°N-26°N, 70°E-90°E that indicates magnitude of the cyclonic shear of the
low-level monsoon circulation.
6) Various studies (Xue et al. 2003; Kripalani et al. 2007; P J et al. 2020; Azhar et al.
2023) highlight a strong dependence of ISM circulation strength on the sea level
pressure difference between the Mascarene High in the Southern Indian Ocean (MH;
20°S-40°S, 45°E-100°E) and the wider ISM region (10°N-35°N, 45°E-100°E).
Therefore, this sea level pressure difference is also treated as an ISM index for this
study.





***2.3.2 Moisture budget calculations***
To understand the mechanisms driving the changes in monsoon precipitation, we
conducted a moisture budget analysis based on the framework of Chou and Lan
(2012). In this analysis, the net precipitation over a region (Precipitation-Evaporation,
P-E) is balanced by the vertically integrated moisture flux convergence in steady
state conditions.

This convergence term is then decomposed into contributions from vertically
integrated horizontal advection (the transport of water vapor, q, by horizontal winds, -
$[u(\partial q/\partial x)+v(\partial q/\partial y)]$) and the transport of moisture by vertical atmospheric motion,
vertical advection ($-[\omega(\partial q/\partial p)]$).

$P-E = -\langle u(\partial q/\partial x)+v(\partial q/\partial y)\rangle - \langle \omega(\partial q/\partial p)\rangle,$                     —------------ (3)

The brackets <> denote pressure, mass-weighted vertical integration. u, v, and ω are
the zonal, meridional, and vertical wind components.
A further decomposition of the advection terms into thermodynamic and dynamic
components to differentiate the contributions from the changes in water vapor and
circulation (e.g., Chou et al. 2009; Chou and Lan 2012) was not performed, as it is
beyond the scope of this study.
**2.4 Decomposition of $\delta^{18}O_{precip}$ changes**
To diagnose the mechanisms driving the changes in precipitation-weighted $\delta^{18}O_{precip}$
in the ISM region between the LGM and PI simulations, we perform a decomposition
analysis following the framework of Tabor et al. (2018). Using our water vapor
tagging results, this method expresses the total change in precipitation-weighted
$\delta^{18}O_{precip}$ ($\delta^{18}O_p$ in the equations below) in the ISM domain as the sum of
contributions from each of the 17 tagged source regions.
$$\Delta\delta^{18}O_p = \sum_{i=1}^{i=17}\left[\left(\delta^{18}O_{p_i}\times\frac{p_i}{p_{total}}\right)_{LGM} - \left(\delta^{18}O_{p_i}\times\frac{p_i}{p_{total}}\right)_{PI}\right]$$         ---------- (4)





286 where δ¹⁸Oₚᵢ is the isotopic ratio of precipitation at the ISM domain of water vapor

287 source i, and (pᵢ/pₜₒₜₐₗ) is the relative contribution of precipitation from source i to the

288 total precipitation at the ISM domain.

289 Further, the decomposition method isolates two primary effects on the change in

290 each δ¹⁸Oₚᵢ between LGM and PI: 1) contributions from changes in the isotopic

291 composition of each source tag between LGM and PI (First term on the right in Eqn.

292 5), and 2) the effect of changes in the relative precipitation contribution from the

293 water vapor sources (Second term on the right in Eqn. 5).

$$\Delta(\delta^{18}O_p)_i = \underbrace{\left(\delta^{18}O_{p_i,LGM} - \delta^{18}O_{p_i,PI}\right) \times \left(\frac{p_i}{p_{total}}\right)_{PI}}_{\text{Change in Isotopic Value}} + \underbrace{\delta^{18}O_{p_i,PI} \times \left(\left(\frac{p_i}{p_{total}}\right)_{LGM} - \left(\frac{p_i}{p_{total}}\right)_{PI}\right)}_{\text{Change in Relative Precipitation Contribution}}$$

294

295 --(5)

296 For each tag, the first term on the right can further be decomposed to three isotopic

297 effects: (i) the source effect, due to changes in δ¹⁸O of water vapor at the source

298 region, (ii) rainout effect, the changes in δ¹⁸Oₚᵢ of source tags due to changes in

299 rainouts on the path, and (iii) condensation effect, the enrichment of δ¹⁸Oₚᵢ at the sink

300 during condensation from the ambient vapor. The three terms are calculated as:

$$\Delta\delta^{18}O_{source,i} = \left(\delta^{18}O_{wv_{source},LGM} - \delta^{18}O_{wv_{source},PI}\right) \times \left(\frac{p_i}{p_{total}}\right)_{PI}$$

301 —-- (6)

302

$$\Delta\delta^{18}O_{rainout,i} = \left[\left(\delta^{18}O_{wv_{sink}} - \delta^{18}O_{wv_{source}}\right)_{LGM} - \left(\delta^{18}O_{wv_{sink}} - \delta^{18}O_{wv_{source}}\right)_{PI}\right] \times \left(\frac{p_i}{p_{total}}\right)_{PI}$$

303

304 - (7)

$$\Delta\delta^{18}O_{condensation,i} = \left[\left(\delta^{18}O_{p_{sink}} - \delta^{18}O_{wv_{sink}}\right)_{LGM} - \left(\delta^{18}O_{p_{sink}} - \delta^{18}O_{wv_{sink}}\right)_{PI}\right] \times \left(\frac{p_i}{p_{total}}\right)_{PI}$$

305

306 -- (8)

307 δ¹⁸Owvsource and δ¹⁸Owvsink are the isotope ratios of water vapor at 850 hPa

308 (representing low level vapor, also level of monsoon low-level jet) of each tag at their

309 source and at the Indian sink, respectively.

310 Hence, we can quantitatively assess the driving mechanisms responsible for the

311 difference in total precipitation δ¹⁸Oprecip between the LGM and PI climates.





**3. Results**
**3.1 PI control simulation**
**3.1.1 Monsoon in the PI control climate**
The model successfully simulates both the annual cycle and the summer monsoon
(mean of June-July-August-September, JJAS) precipitation, also the south westerly
winds over the ISM domain (8°N-30°N and 65°E-88°E; Fig. 1). The summer
monsoon precipitation accounts for approximately 80% of the total annual
precipitation, in agreement with the GPCP observational data (Adler et al. 2018).
However, the iCESM overestimates the summer monsoon precipitation by ~20%
(Fig. 1c). This wet bias has been reported in previous studies that used the CESM
model (e.g. Pathak et al. 2019; Hanf and Annamalai 2020) and they attribute this
bias to factors such as, model resolution, biases in simulated circulation, and
convective parameterizations.

**3.1.2 Water isotopes in the PI monsoon precipitation**
The domain mean water isotope ratio of precipitation (precipitation weighted,
$\delta^{18}O_{precip,}$ Fig. 2a) over the ISM region during the JJAS season in the PI simulation is
-7‰. This is considerably more negative than the mean of -3.7‰ calculated from the
available GNIP observational data over the domain (Fig. 2a). A likely reason for the
more negative $\delta^{18}O_{precip}$ values simulated in the ISM region is the wet bias in the
model (Fig. 1c) and consequent depletion of the heavier isotopes, as also suggested
by previous studies using isotope-enabled CAM and iCESM models (Nusbaumer et
al. 2017; Tharammal et al. 2017; Tharammal et al. 2023). It should be noted,
however, that a lack of wider observational networks and continuous monitoring of
seasonal $\delta^{18}O_{precip}$ values hinder a comprehensive comparison of the observations
and our simulation.

In the PI simulation, the linear regression analysis between the JJAS mean $\delta^{18}O_{precip}$
values and the precipitation amount show a moderate amount effect in the ISM
region (-0.22‰/mm day$^{-1}$ slope of the spatial amount effect, the square of the
Pearson correlation coefficient $r^2$ 0.37, Fig. S3). The moderate strength of this





relationship, which is physically related to rainout during heavy rainfall and
convective events (Risi et al. 2008; Tharammal et al. 2017; Lee and Fung 2008),
suggests that factors other than local precipitation amount also strongly influence the
simulated $\delta^{18}O_{precip}$ values. These may include large-scale circulation, upstream
convection, or the effects of water vapor sources with differing isotope signatures
(Pausata et al., 2011; Risi et al., 2008; Tharammal et al., 2023), as discussed in the
following section.
**3.1.3 Water vapor sources and their effect on $\delta^{18}O_{precip}$ in the PI climate**

We used source water tagging to identify the primary water vapor sources for ISM
precipitation in the PI simulation. Our simulation shows that the 17 tagged source
regions (Fig. S2) contribute approximately 96% of the total JJAS precipitation (Fig.
2b). Four major sources- the South Indian Ocean (SIO) and Central Indian Ocean-
CIO (22% and 10% respectively), Arabian Sea (19%), and Indian land recycling
(17%), together account for ~68% of the total precipitation. The Bay of Bengal (BOB)
contributes only ~3% to the ISM precipitation. These results are consistent with
previous water vapor tracking studies in the ISM domain using Lagrangian models
(e.g. Gimeno et al. 2010, 2012; Ordóñez et al. 2012; Pathak et al. 2014; Dey and
Döös 2021) and present-day results using the iCESM model (Tharammal et al.

362    2023).


The precipitation contribution-weighted sum of $\delta^{18}O_{pi}$ of all the 17 tags at the sink (-
6.7‰, based on Eqn. 4) explains ~95% of the domain mean $\delta^{18}O_{precip}$ in the ISM
region (-7‰, cf. 3.1.2), which validates our source-tagging framework (Eqn. 4, Fig.
2c). The results show substantial differences in the isotope signatures between the
major sources, mainly influenced by transport distance. For instance, while the
Arabian Sea and SIO contribute comparably to JJAS precipitation (19% and 22%,
respectively), their water isotopic signatures in precipitation ($\delta^{18}O_{pi}$) greatly differ.
The Arabian Sea source is relatively enriched (-0.1‰ mean over the ISM region),
whereas the SIO has much depleted $\delta^{18}O_{pi}$ values of -2.5‰ (Fig. 2c). This is likely
due to the larger distance of the SIO source from the ISM sink region and
consequent depletion of the vapor during condensation and rainouts in the path.
Similarly, the evapotranspiration from the ISM land domain, recycling source,



contributes 17% to the total precipitation, and the $\delta^{18}O_{pi}$ values of recycling are
comparatively enriched (-0.6‰), likely due to being the local source of vapor. Hence,
we suggest that the isotopic composition of ISM precipitation is sensitive to the
relative contributions of these dominant water vapor sources and their isotopic
signatures.

**3.2 Global climate response in the LGM simulation**
In the LGM simulation, the annual global mean surface temperature cooled by
6.75°C compared to the PI (Fig. S4a). While this cooling is consistent with coupled
CESM simulations (-6.8°C; Zhu et al. 2017; Tierney et al. 2020), it is greater in
magnitude than the PMIP4 multi-model mean (Kageyama et al. 2020). The cooling is
more pronounced over the Laurentide ice sheets and in the polar regions, due to ice
sheet albedo feedback and polar amplification. This leads to an asymmetry in the
annual cooling between the two hemispheres, with Northern Hemisphere (NH)
cooling (-7.5°C) exceeding that of the Southern Hemisphere (SH; -6.0°C)
($\Delta TS=1.4$°C). This interhemispheric asymmetry in cooling is smaller than previous
modeling studies that found values more than 3°C (Broccoli, 2000). The simulated
cooling in the high latitude ocean regions (~5 to 6°C, Fig. S1a) agrees well with the
proxy-reconstructions (MARGO Project Members 2009). However, the model
simulates colder SSTs in the tropics compared to the MARGO and GLOMAP
reconstructions as noted by previous studies (~3°C in CESM simulations versus
~1.5°C; (Tierney et al. 2020).

Globally, the annual mean precipitation reduces by ~12% (Fig. S4b) in the LGM,
consistent with proxy records and modeling studies including the PMIP4 LGM
simulations (Bartlein et al. 2011; Yan et al. 2016; DiNezio et al. 2018; Kageyama et
al. 2020). This reduction corresponds to a global hydrological sensitivity of ~1.8% per
°C of cooling, and is close to the estimated thermodynamic increase in global
precipitation of 2% per unit increase in temperature (Trenberth 2011). Despite the
reduction in global mean precipitation, an increase in precipitation is simulated in
some regions such as, tropical Pacific, parts of N. America, and South Africa, and
these patterns are also found in the PMIP4 simulations (Kageyama et al. 2020).






Furthermore, we find that the position of the annual mean Intertropical Convergence
Zone (ITCZ), defined as the median of zonal precipitation (20°S-20°N; McGee et al.
2014; Devaraju et al. 2015), shifts northward by 1.2° in the LGM simulation. This
finding contrasts with the southward shift of ITCZ reported in many of the PMIP4
models (Wang et al. 2023), but is consistent with results from CESM2 (Zhu et al.
2022; Lofverstrom and Zhu 2023) as also reported in Wang et al. (2023).
Lofverstrom and Zhu (2023) attribute this possible bias in the LGM ITCZ shift to
biases in the model's cloud microphysics. It is likely that the simulated northward
displacement of the ITCZ in the LGM in our simulation is due to a robust increase in
precipitation over the northern tropical Pacific, coupled with widespread drying in the
Southern Hemisphere tropical regions (Fig. S4b).
**3.3 Indian summer monsoon precipitation during the LGM**
The LGM simulation shows a substantial reduction in the Indian summer monsoon
precipitation (Fig. 3), characterized by widespread drying over India, SE Asia, and
Arabian Peninsula region. The precipitation amount is reduced by ~15% over the
ISM domain, which is notable since a precipitation deficit exceeding 10% from the
long-term mean is considered drought conditions in India (Shewale and Kumar
2005). The ISM precipitation responses in the LGM simulation are broadly consistent
with both monsoon proxy-records and previous climate model simulations (Jiang et
al. 2015; Dutt et al., 2015; Yan et al. 2016; Kageyama et al. 2020). The large-scale
drying is primarily due to regional and global cooling in both annual and summer
means (Fig. S4a, S5a) and generally reduced evaporation from the tropical oceans
(Fig. S5b). These patterns are linked to decreased atmospheric humidity and
reduced column-integrated precipitable water (reduction of 25.2% over the ISM
domain, Fig. S6). However, the drying during the summer monsoon season is not
uniform across the region. Increased precipitation is simulated in the east part of
India and the Bay of Bengal (Fig. 3). As this increase cannot be explained by the
precipitable water anomalies in the LGM (Fig. S6), it is likely driven by changes in
atmospheric circulation.



### 3.3.1 Monsoon circulation changes in the LGM and moisture budget


The LGM simulation shows a weakening of the low-level (850 hPa) westerly
circulation and wind speeds towards land over the Northern Arabian Sea (Fig. 3, Fig.
S7b), driven by substantially weakened land-ocean thermal (Fig. S5a, larger cooling
over the land) and pressure gradients (S8b; Roxy et al. 2015; Weldeab et al. 2022).
Surface cooling over the Indian subcontinent (domain mean -4.5°C; Table S2) in the
LGM is approximately 1°C greater than the sea surface temperature cooling in the
neighbouring Arabian Sea, which is consistent with the lower heat capacity of land,
leading to more pronounced cooling and enhanced surface pressure over land.

The circulation response is not uniform across the region, as a regional
intensification of the low-level westerly winds is simulated across the central and
southern parts of India and the Bay of Bengal (Fig. 3, Fig. S7b). This regional
intensification of the monsoon circulation is captured by several monsoon circulation
indices used in this study- an increased vertical shear of zonal winds, strengthening
of the Somali Jet, and enhanced barotropic shear (Fig. S9b, d, f, respectively). We
suggest the enhanced westerly circulation in parts of the monsoon region, especially
the Somali jet, is influenced by a stronger Mascarene high in the Southern Indian
Ocean (Fig. S9e, S8b) that enhances the pressure gradient between the Indian land
and the Southern Indian Ocean by ~2 mb (Fig. S8b, S9e). The strengthened
Mascarene high is a result of the sea ice extension and cooling in the Southern
Indian Ocean during the LGM (Fig. S1b, c). This is in agreement with the positive
relationship between the ISM circulation and pressure gradient between the Indian
monsoon region and the Mascarene high, suggested by several previous studies
(Kripalani et al. 2007; P J et al. 2020; Azhar et al. 2023). However, the tropospheric
temperature gradient (ΔTT), shows a weakening by 2.5% in the LGM. This indicates
a weaker thermal forcing of the monsoon, likely due to enhanced cooling in the
northern box used for the estimation of ΔTT (Fig. S2b, S5a), in the LGM simulation.

We note that the indices related to monsoon circulation (vertical shear of zonal
winds, Somali jet speed index, pressure gradient between MH and ISM regions that
characterize dynamical responses (Fig. S9) show a general strengthening of the
monsoon circulation by ~12-15% in the LGM simulation, compared to the PI. The





barotropic index shows an even larger percentage change between LGM and PI of

>100%. However, the index related to water vapor content and its transport (the
monsoon hydrological index and Vertically Integrated Moisture Transport VIMT that
characterize thermodynamical responses, (Fig.  S9c, S10, Fasullo et al. 2003) shows
a reduction by 7.5%, along with a reduction of column-integrated precipitable water
over the ISM region by ~25% (Fig. S6). This shows that the reduction in the ISM
precipitation in the LGM simulation is mainly due to the thermodynamic response to
the cooling (reduced water vapor in the atmosphere), despite an enhanced
dynamical response (circulation changes).

To understand the drivers of the regional precipitation changes, we analysed the
surface moisture budget (net precipitation, P-E), decomposing it into contributions
from horizontal and vertical advection of moisture (Section 2.3, Chou and Lan,
2012). The analysis (Fig. 4) shows that the major term related to the drying (Fig. 4a)
in the ISM region is the decrease in the horizontal moisture advection (Fig. 4b). This
is due to both the reduction in the moisture availability, and reduced transport as
discussed before. In contrast, the increased precipitation in eastern part of India and
BOB in the LGM is caused by enhanced moisture convergence and vertical
advection (Fig. 4c) linked to the intensified monsoon westerlies in that region. We
note that these results for the advection terms include both dynamic and
thermodynamic responses (Chou and Lan 2012) and delineating them is out of the
scope of this paper.

The ISM precipitation reductions are also associated with large-scale zonal
temperature gradients between a cold tropical western Pacific Ocean and a relatively
colder Indian subcontinent (Fig. S5a). This leads to anomalous updrafts over the
western Pacific, and increased subsidence over the Indian region (Fig. 5b, Fig.
S11b, d). The relationship between a warmer W. Pacific and drying over the Indian
region is discussed in previous studies (Annamalai et al. 2013). Furthermore, the
Western tropical Pacific is ~1.5°C warmer compared to the Central and Eastern
tropical Pacific in the LGM simulation (Fig. S5a). This intensifies the Pacific Walker
circulation further, and enhances the subsidence and drying over the Eastern Pacific
and the Indian subcontinent in the LGM. We suggest this large-scale circulation
response enhanced the drying over India in the LGM simulation.





**3.4 Monsoon water vapor sources under glacial conditions**

In the LGM simulation, the major four sources - SIO, Arabian Sea, recycling, and CIO- remain unchanged (23%, 21%, 19%, and 10% contributions to total precipitation, respectively) and their relative contributions change by less than 4% compared to the PI (Fig. 6b). However, the absolute amount of moisture from each source decreases by 10-14% (Fig. 6a). This reduction is primarily driven by reduced evaporative fluxes over the source regions (up to ~50% from the PI values; Fig. S5b, Table S2) and a general weakening of the moisture transport (Fig. S8b). The reduction in horizontal advection term over the ISM region in the moisture budget (Fig. 4b) corroborates with these results. This suggests that changes in atmospheric circulation and cooling of sea surface temperatures during the LGM significantly impacted the availability and transport of moisture to the Indian monsoon region.

**3.5 $\delta^{18}O_{precip}$ in the LGM**

Globally, the LGM simulation shows a strong depletion in annual mean $\delta^{18}O_{precip}$ values (by 5 to >10‰) over the high latitudes and continental ice sheets (Fig. S4c). This is mainly due to the "temperature effect", as the cooling in the LGM leads to a stronger Rayleigh distillation process (Galewsky et al. 2016). Previous studies (Broccoli and Manabe 2008; Tharammal et al. 2013; Zhu and Poulsen 2021; Kageyama et al. 2020) have shown that reduced GHG and consequent cooling, changes in circulation, and both topography and albedo of the ice sheets contribute to this depletion in the high latitudes. In the following text, we discuss the changes in $\delta^{18}O_{precip}$ over the ISM region during the JJAS season.

In contrast to the high latitudes, considerable enrichment of $\delta^{18}O_{precip}$ (1‰-4‰) over tropical regions including the Indian Ocean, Southeast Asia, and ISM regions (mean enrichment of 0.9‰ over the ISM domain) is simulated in the JJAS season (Fig. 7). This simulated enrichment is in agreement with the proxy data records from the South Asian summer monsoon region and climate model simulations (Hoffmann and Heimann 1997; Tiwari et al. 2011; Liu et al. 2014; Jiang et al. 2015; Kathayat et al. 2016; Kaushal et al. 2018).





The JJAS mean amount effect in the LGM (spatial slope -0.24 ‰/mmd$^{-1}$, r²=0.30,
Fig. S3) is moderate and similar to that in the PI simulation. Importantly, the linear
regression analysis (Fig. S3) shows that there is no significant correlation between
the changes in precipitation (ΔP) and the changes in $\delta^{18}O_{precip}$ between the LGM and
PI simulations (temporal slope of amount effect, $\Delta\delta^{18}O_{precip}/\Delta P$, slope -0.09
‰/mmd$^{-1}$, r²=0.07). Hence, the LGM enrichment in the ISM region cannot be
explained by the amount effect, and the results indicate the influence of other factors
such as changes in water vapor sources and atmospheric circulation.

**3.6 Drivers of monsoon $\delta^{18}O_{precip}$ changes: Perspectives from source tagging**
To diagnose the physical processes responsible for the changes in the monsoon
$\delta^{18}O_{precip}$ during the LGM, we conducted a decomposition analysis of the JJAS mean
LGM-PI $\delta^{18}O_{precip}$ anomalies, following (Tabor et al. 2018). Details of the calculations
are given in Section 2.4. Using the results from water tagging experiments, we
separated the anomalies (LGM-PI) in the $\delta^{18}O_{precip}$ of each source tag ($\Delta\delta^{18}O_{pi}$) into
4 components: 1) effects of changes in source vapor $\delta^{18}O$ ($\Delta\delta^{18}O_{source}$), 2) effects of
changes in rainout during transport ($\Delta\delta^{18}O_{rainout}$), 3) effects of changes in
condensation over the Indian monsoon domain ($\Delta\delta^{18}O_{condensation}$), and, 4) effects of
changes in the relative contributions of each source to total precipitation over India
(Shown in Fig. 8a-d).


The analysis shows that the dominant contributor to the positive anomalies in the
$\delta^{18}O_{precip}$ values over the ISM domain is the change in the relative contribution of the
water vapor sources, which accounts for an enrichment of +0.6‰ (Fig. 8d). This is
caused by a reduction in the relative contribution from remote and depleted water
vapor sources- North and South Pacific, Atlantic, and South China Sea (Fig. 8d).
The second largest positive contribution is from the Rainout Effect (+0.4‰ in total,
Fig. 8b). This is driven by a weaker rainout along transport pathways from major
water vapor sources -Southern and Central Indian Ocean, due to a weaker
circulation in many parts of the Indian Ocean, and overall reduced rainfall in the
LGM. In contrast, the Source Effect (effects of changes in source vapor $\delta^{18}O$, Fig.
8a) provides a small net negative contribution, as a positive contribution from the



Arabian Sea is offset by negative effects from other source regions (Fig. 8a). The
positive effect from the Arabian Sea source is likely due to a localized increase in
evaporation (Fig. S5b) in contrast to other sources where evaporation was generally
reduced, and also the prescribed 1‰ global ocean surface enrichment in the LGM
simulation.

The condensation term, which represents the local enrichment of the precipitation at
the sink during the phase transition of vapor to precipitation, produces only a minor
and slightly negative contribution (Fig. 8c). This suggests that the isotopic
enrichment of precipitation on condensation was weaker in the LGM compared to the
PI. This finding also confirms that the amount effect is not a primary driver of the
LGM enrichment. If a strong amount effect existed, the reduced LGM precipitation
should have produced a positive condensation term. The negative contribution from
the condensation term, therefore, agrees with our previous analysis showing no
significant temporal correlation between changes in ISM precipitation and isotopes
(Section 3.5; Figure S3) in the LGM.
**4. Discussion and Conclusions**
The present study used a water isotope, water tagging-enabled general circulation
model to investigate the Indian summer monsoon precipitation and isotope
responses under glacial conditions. Our simulations show a 15% reduction of
monsoon precipitation over the Indian domain. Our study shows that the reduction in
Indian monsoon precipitation is due to the effects of global cooling and reduced
humidity (due to reduced $CO_2$ and the presence of continental ice-sheets;
(Kageyama et al. 2020), and a weakened land-ocean temperature gradient which
reduced the strength of the monsoonal circulation in many parts of the ISM region.
The LGM drying over the Indian subcontinent was enhanced by a Walker-like
circulation response, driven by zonal temperature gradients between the less-cooled
Western Pacific and the cooler Indian subcontinent, which created anomalous
subsidence over the Indian region.
The reduction in the summer monsoon precipitation in the LGM simulation is
consistent with climate models and proxy records of monsoon precipitation (Liu et al.





2021; Jiang et al. 2015; Wang et al. 2023; Yan et al. 2016; Cao et al. 2019). The
simulated northward shift of the ITCZ in our iCESM results, likely due to increased
tropical North Pacific precipitation, conflicts with the southward shift simulated by
several other models (Wang et al. 2023). This discrepancy points to uncertainties in
climate simulations and suggests that more studies are required to assess the
representation of tropical ocean-atmosphere interactions under the glacial climate
conditions.
We also note that the low-level circulation responses in the LGM simulation
(enhanced cyclonic barotropic shear with enhanced westerly anomalies over
Southern India, and easterlies over the northern latitudes) is consistent with the
climate model responses in future warming scenarios (Menon et al. 2013). Menon et
al. (2013) find that under the RCP8.5 scenario, CMIP5 models project a weaker low
level cyclonic monsoon circulation with enhanced westerly anomaly over northern
India and easterly anomaly over the south, despite a simulated increase in the
monsoon precipitation. Thus, our results are consistent with monsoon responses in
future warming scenarios, such that in the colder LGM conditions, the monsoon
precipitation is reduced due to thermodynamic response to cooling, while the
dynamical response characterized by monsoon circulation indices in general is
intensified.
A key contribution of this study is the novel application of water vapor source-
tagging to the LGM climate simulation, which shows that the major water vapor
sources for the Indian monsoon were unchanged between the pre-industrial and
LGM climates. Our study finds that isotopic ratio of precipitation is enriched by ~1‰
over the ISM domain during the LGM, which is in agreement with Speleothem proxy
records from Mawmluh (Dutt et al. 2015) and Bittoo caves (Kathayat et al. 2016) in
North India. Our analysis confirms the amount effect (Dansgaard 1964) was not the
primary driver for this enrichment, as we find no significant correlation between the
changes in precipitation and $\delta^{18}O_{precip}$ from the PI to the LGM. Instead, our
decomposition analysis using water vapor tagging finds that the simulated LGM
enrichment is due to a reduced relative contribution from distant, isotopically
depleted moisture sources and decreased rainout from Indian Ocean sources. The
results are in agreement with studies by Tabor et al. (2018) and Hu et al. (2019) who



find the importance of different water vapor sources for the South Asian and East
Asian monsoon $\delta^{18}O_{precip}$ values. Hence, this study emphasizes that rather than
being a simple proxy for local rainfall, the $\delta^{18}O_{precip}$ values in the ISM region is a
complex signal integrating large-scale atmospheric dynamics and moisture sources.
Further, these findings pose questions for the interpretation of paleoclimate records
where $\delta^{18}O$ values of climate records are used as a direct proxy for local
precipitation intensity.
We note that our study has a few limitations. The version of the CESM model we
used has high climate sensitivity and overestimates the LGM global cooling (Zhu et
al. 2022), an issue attributed to its cloud parameterization. However, we suggest that
these biases do not affect our key results, as the model is able to both successfully
capture the present-day monsoon circulation, and isotopic distribution. The model is
also able to simulate the isotopic enrichment found in proxy records during the LGM.
Further, we use a single model, with prescribed SSTs and prescribed surface ocean
water isotope ratios because of the cost of computation when we utilize the water-
tagging capabilities of the model. Future work should employ fully coupled Earth
system models within a multi-model framework to investigate ocean-atmosphere-
isotope feedback and test the robustness of these results.
In conclusion, this study disentangles the drivers of Indian monsoon precipitation and
its isotopic signature during the LGM. The results highlight that the isotopic
composition of precipitation in the Indian monsoon region is a complex signal
integrating changes in circulation, changes in relative contribution of water vapor
sources, and upstream rainout processes. These findings underscore the importance
of considering moisture source and transport history when interpreting paleoclimate
isotope records from the Indian monsoon and other tropical monsoon regions.
**Code/Data availability**
The datasets used in the current study will be made available to the public.
**Author contributions**



TT: Conceptualization, Funding acquisition, Methodology, Investigation, Formal
analysis, Visualization, Writing - Original Draft. GB: Methodology, Analysis, Writing -
Review & Editing. JN: Methodology, Model software, Analysis, Writing - Review &
Editing.
**Competing interests**
None of the authors has any competing interests.
**Acknowledgements**
TT is supported by the DST-INSPIRE Faculty Fellowship awarded by the
Department of Science and Technology, India and Anusandhan National Research
Foundation (ANRF) Early Career Research Grant. We acknowledge Jiang Zhu
(NCAR) for providing the boundary conditions for the simulations, and we would like
to thank Dr. André Paul for discussions during the initial stages of this research. We
acknowledge the high-performance computing support from the Supercomputer
Education and Research Centre (SERC), Indian Institute of Science, Bangalore. All
the figures in the manuscript were created with NCL (NCAR Command Language
Version 6.6.2, http://www.ncl.ucar.edu/).

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

**Main Figures**



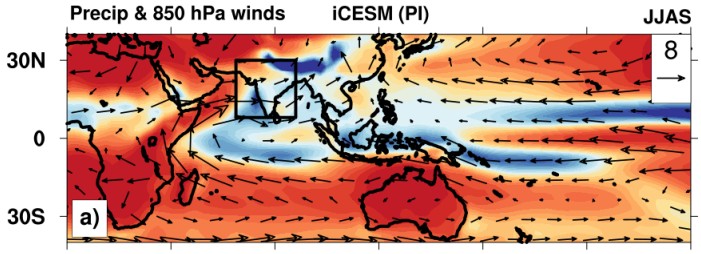

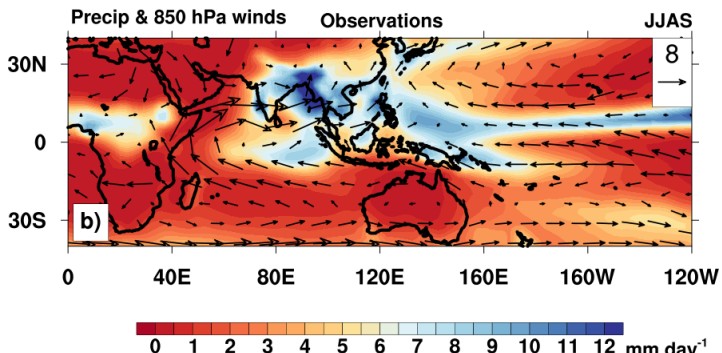

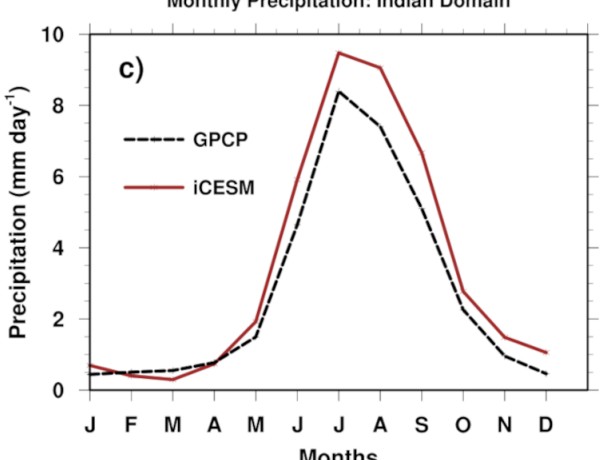


Figure 1: Model-simulated and observed precipitation (shaded, mm day⁻¹) and 850
hPa winds (vectors in m s⁻¹, reference vectors are shown in the panels) for the
summer monsoon season (mean of June-July-August-September-JJAS). Panel (a)
shows the simulated JJAS mean precipitation and 850 hPa winds from the iCESM
Pre-Industrial (PI) simulation. Panel (b) shows the corresponding JJAS long-term
mean precipitation from GPCP (Adler et al., 2018) and 850 hPa winds from ERA5
reanalysis (1980-2000; Hersbach et al., 2020). Panel (c) shows the monthly mean



precipitation averaged over the land grid cells in the Indian domain (8°N-30°N, 65°E-
88°E; black box in panel a, comparing the iCESM simulation with GPCP
observations.










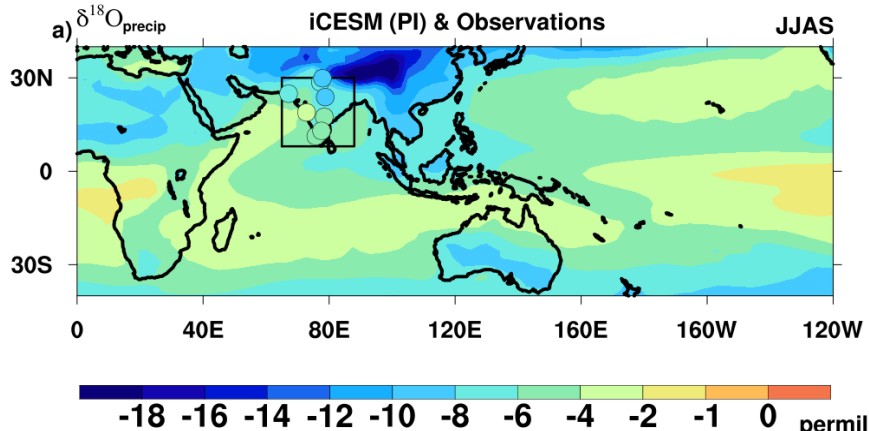

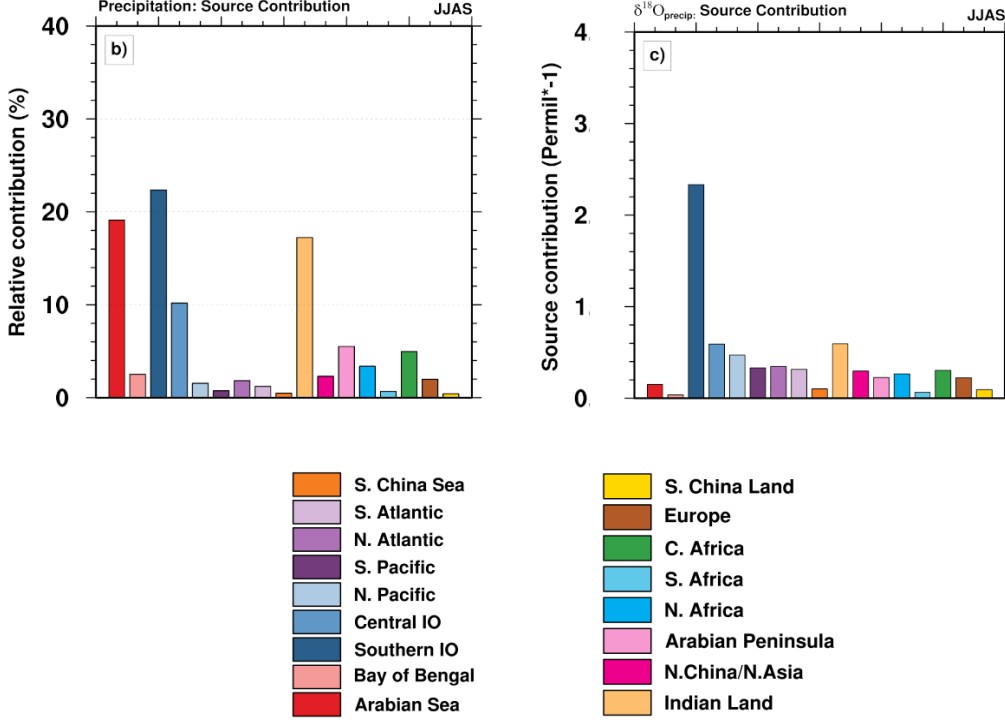

Figure 2: Isotopic composition of precipitation (δ¹⁸Oprecip) for the Indian summer


monsoon season in the pre-industrial (PI) simulation, shown along with relative

contribution of precipitation from the tagged sources, and their δ¹⁸O values in

precipitation (weighted by relative contribution of precipitation).

Panel (a) shows the mean JJAS δ¹⁸Oprecip (shading, in permil [‰]). The filled circles

in the Indian domain (8°N-30°N, 65°E-88°E; shown in black box in panel a) represent





long-term JJAS mean observational data from Global Network of Isotopes in
Precipitation (GNIP) stations. Panel (b) shows the relative contribution ($P_{tag}/P_{total}$, in
%) of precipitation to the Indian summer monsoon domain from 17 tagged water
vapor source regions. Panel (c) shows the $\delta^{18}O_{precip}$ of tagged precipitation from the
17 different source regions that contribute to the Indian monsoon precipitation. The
y-axis values in panel (c) are multiplied by -1 for visualization purposes (units of -‰).



























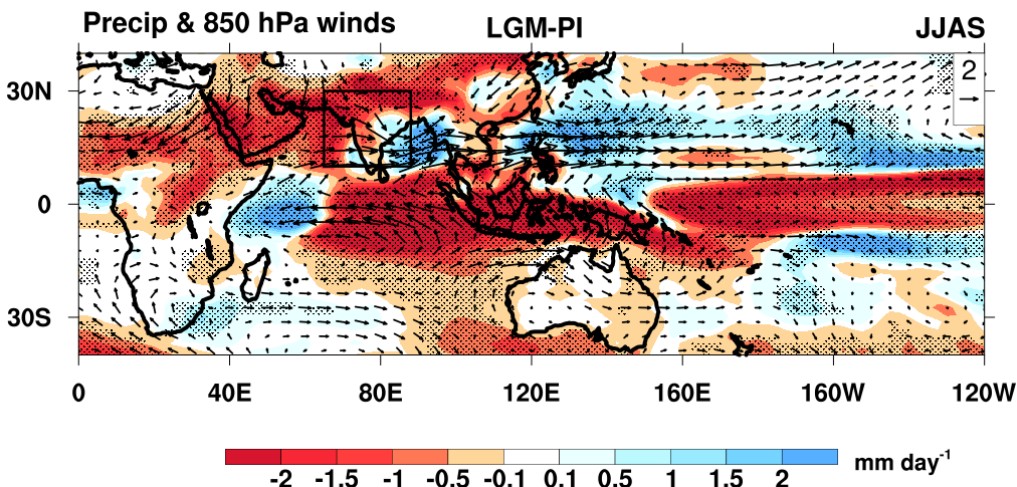

Figure 3: The simulated differences in JJAS mean precipitation (shaded, in mm day$^{-1}$) and low-level winds (850 hPa, vectors shown in m s$^{-1}$) between the pre-industrial and the Last Glacial Maximum (LGM) simulations, shown as LGM-PI. Regions where the anomalies are statistically significant at the 95% confidence level are stippled. Significance level is estimated using a Student's t test from a sample of 20 annual means from the control and LGM simulations.



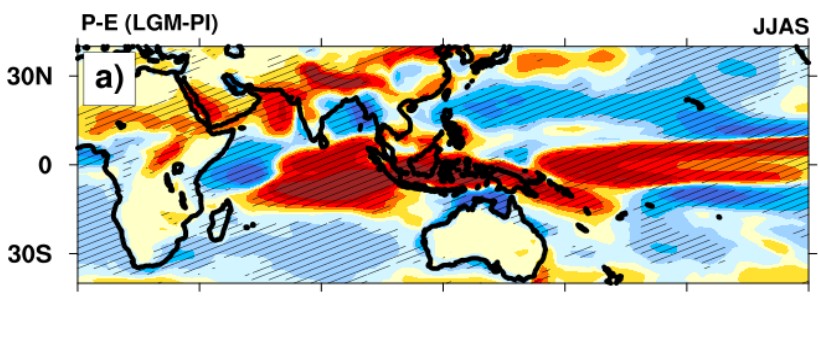

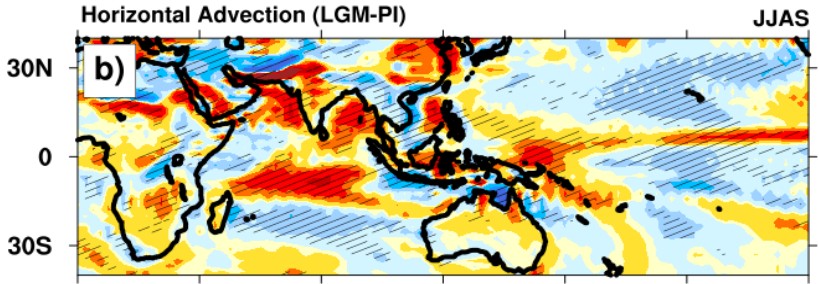

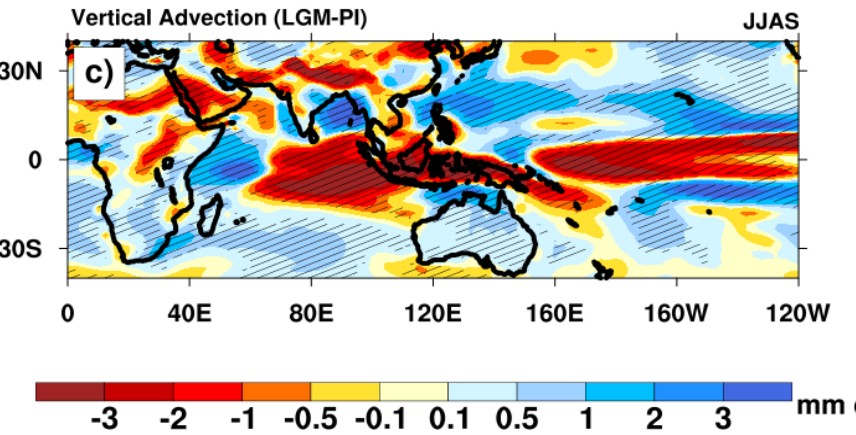

Figure 4: Changes in the JJAS mean atmospheric moisture budget between the
LGM and the PI simulations, shown as LGM-PI. The panels show the components of
the vertically integrated moisture budget anomaly: Panel a) shows anomalies in
precipitation minus evaporation (P-E). Panel b) shows anomaly in horizontal water
vapor advection ($-\langle v \cdot \nabla q \rangle$). Panel c) shows anomaly in vertical water vapor
advection ($-\langle \omega \partial q / \partial p \rangle$). V is the horizontal wind, q specific humidity, p atmospheric
pressure, and ω pressure velocity. All panels have the units of mm day⁻¹. The



hatching shows regions where the anomalies are statistically significant at the 95%
confidence level. Significance level is estimated using a Student's t test from a
sample of 20 annual means from the control and LGM simulations.
















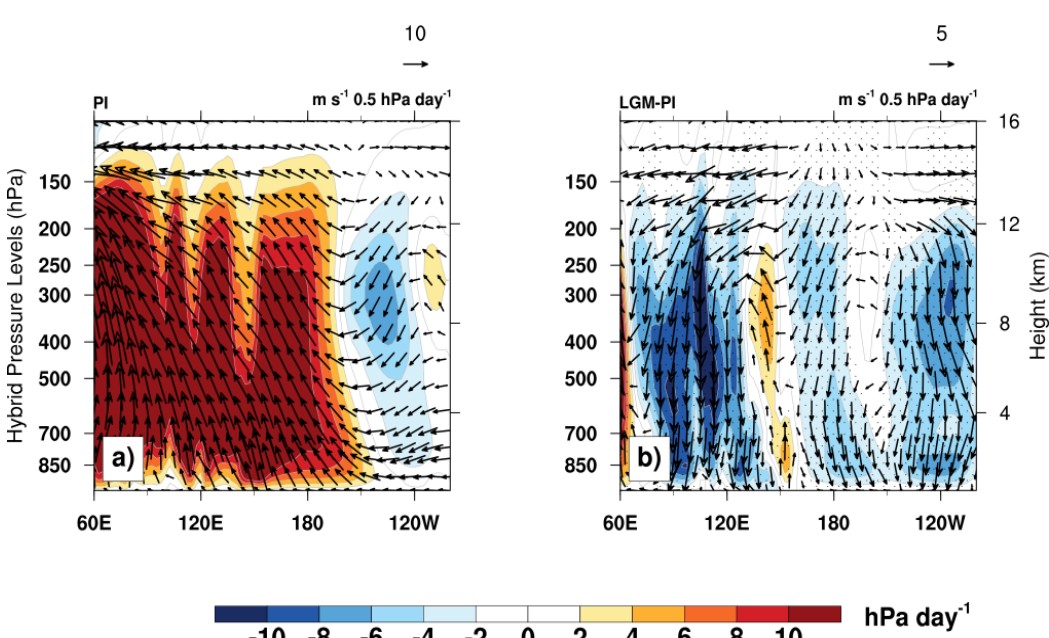

Figure 5: The tropical zonal circulation during the JJAS season, averaged between 10°S and 10°N. Panel a) shows the circulation in the PI control simulation, and the right panel b) shows the anomalies between the LGM and PI simulations (as LGM-PI). In both panels, shading represents the vertical pressure velocity (-ω), where blue shading indicates downward motion and red shading indicates upward motion. The vectors show the zonal-vertical circulation, composed of zonal wind (u, in m s⁻¹) and vertical pressure velocity (-ω), with the ω scaled by 0.5 hPa day⁻¹ for visualization. The reference vectors are shown in the top right of the panels. The stippling in panel b) shows regions where the anomalies are statistically significant at the 95% confidence level. Significance level is estimated using a Student's t test from a sample of 20 annual means from the control and LGM simulations.











Figure 6: Changes in the precipitation contribution from 17 tagged moisture source
regions to the Indian monsoon domain's JJAS mean precipitation between the LGM
and the PI simulations. The source regions corresponding to each bar are identified
in the legend. Panel (a) shows the absolute difference in precipitation contribution
from each source region ($\Delta P_{tag}$). The values are shown in mm day$^{-1}$ and have been
scaled up by a factor of 10 for visualization. Panel (b) shows the difference in the
relative contribution of each source to the total precipitation at the Indian monsoon
domain ($\Delta[P_{tag}/P_{total}]$), shown as a percentage (%).




















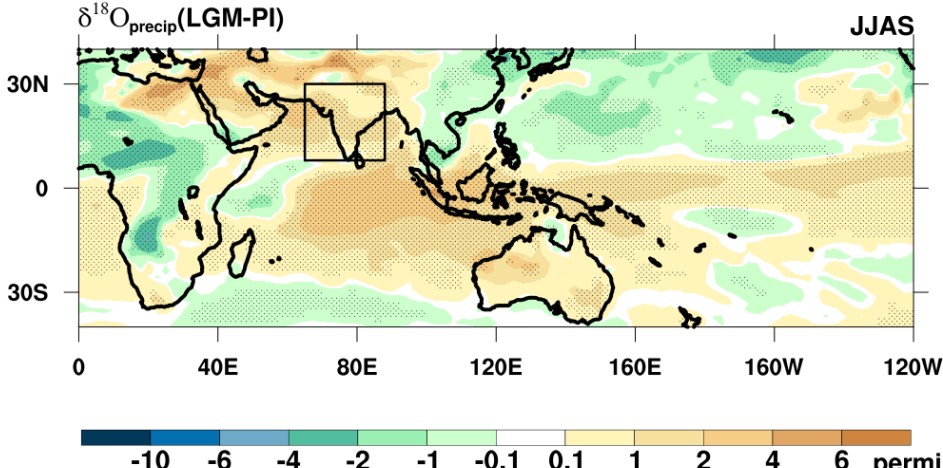


Figure 7: The anomalies of JJAS mean precipitation weighted $\delta^{18}O_{precip}$ in permil between the LGM and PI simulations as LGM-PI. The Indian monsoon domain is shown in a black box in the plot. Regions where the anomalies are statistically significant at the 95% confidence level are stippled. Significance level is estimated using a Student's t test from a sample of 20 annual means from the PI control and LGM simulations.
















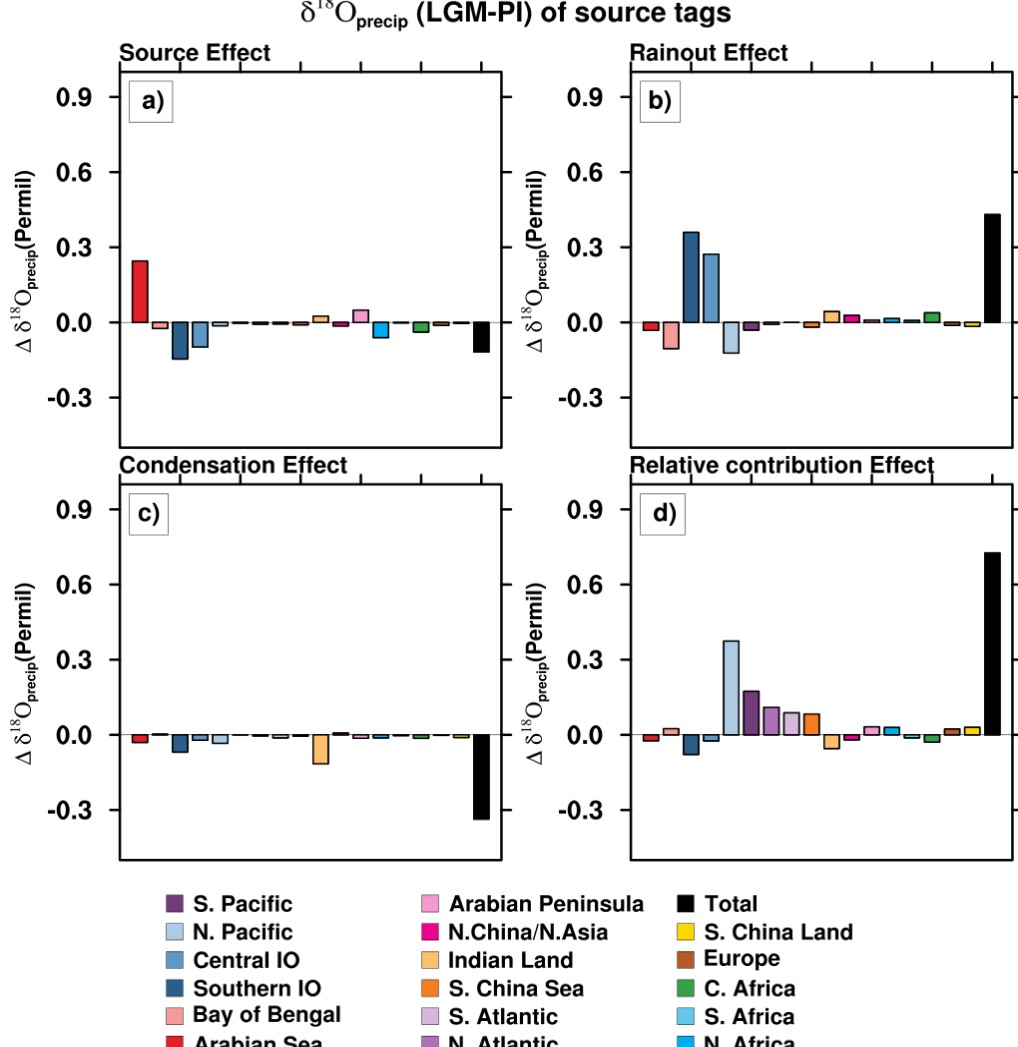


Figure 8: Decomposition of the change in JJAS mean precipitation $\delta^{18}O$ ($\Delta\delta^{18}O_{precip}$) for the Indian monsoon domain between the LGM and PI simulations. All values are in permil (‰). The $\Delta\delta^{18}O_{precip}$ is divided into contributions from 17 tagged moisture source regions, shown in the legend. The decomposition separates the total anomaly into four primary physical processes for each tag. Panel a) shows the Source Effect: Changes in the $\delta^{18}O$ of water vapor at its evaporative source. Panel b) shows the



Rainout Effect: Changes in isotopic composition due to rainout during atmospheric
transport from the source region to the ISM domain. Panel c) shows the
Condensation Effect: Changes in the isotopic fractionation during the conversion of
water vapor to precipitation over the monsoon region. Panel d) shows the
Precipitation Relative Contribution Effect: Changes in $\delta^{18}O_{precip}$ for each tag resulting
from shifts in the relative contribution of precipitation from different source regions.



