# Peer review of "Drivers of the $\delta^{18}O$ Changes in Indian Summer Monsoon Precipitation between the Last Glacial Maximum and Pre-industrial Period Thejna Tharammal 1\*, Govindasamy Bala2, Jesse Nusbaumer3 1 Interdisciplinary Centre for Water Research, Indian Institute of Scie"

_EGUsphere, 2025_

## Author Comment (AC1)

**Comments and Responses on manuscript egusphere-2025-4468: Reviewer 1**

This study examines changes in the isotopic composition of Indian summer monsoon precipitation during the Last Glacial Maximum (LGM) relative to the pre-industrial period, using an isotope-enabled general circulation model with vapor source tagging. The authors found that the LGM simulation shows 15% less monsoon rainfall, mainly due to thermodynamic drying from lower atmospheric moisture and enhanced subsidence over India. With the water vapor source tagging method, they further found that while primary moisture sources remain the same, their contributions weaken, producing $\delta^{18}O$ enrichment. This enrichment stems mainly from reduced input of isotopically depleted vapor rather than the local amount effect. The results suggest that $\delta^{18}O$ in Indian monsoon records reflects large-scale circulation changes rather than local precipitation intensity. The paper is very well organized and clearly described. I would like to recommend an acceptation for publication after some minor improvements.

We thank the referee for the thoughtful and constructive comments, as well as for the positive recommendation for publication of our manuscript. Below are our responses to their comments and suggestions. The reviewer's comments are shown in black, and our responses are in blue.

For Fig. 4, it would be better to also mark the research domain as in Fig. 3.

Thank you for the suggestion. We have now marked the Indian monsoon domain in Figure 4.

1. One of the main points of this study is to explain the reason of the reduced rainfall during the LGM compared to the pre-industrial period. While the manuscript has only 8 figures in the main text, I suggest to change one Figure (Fig. S8 or S9) from the supplements into the main text to show the corresponding changes in temperature and circulation.

Thank you for the suggestion. We agree that presenting circulation changes in the main text will help to increase clarity. Accordingly, we will move Fig. S9 (Monsoon Circulation Indices) from the Supplement to the main text.

---

## Author Comment (AC2)

**Comments and Responses on manuscript egusphere-2025-4468: Reviewer 2**

This manuscript uses isotope-enabled CESM with water-vapor source tagging to compare Indian Summer Monsoon precipitation and δ18O between the Last Glacial Maximum and pre-industrial climates. The authors find monsoon drying at LGM and enrichment of precipitation d18O, and convincingly show that the enrichment cannot be explained by the local amount effect but instead arises primarily from reduced relative contributions from distant, isotopically depleted sources and weaker upstream rainout along transport pathways. The modeling framework and decomposition are solid, and the results are clearly presented. I recommend minor revision.

We appreciate the time and effort that the reviewer devoted to our manuscript, and we thank the reviewer for the detailed and constructive assessment of our work. Below are our responses to their comments and suggestions. The reviewer's comments are shown in black, and our responses are in blue.

Minor comments:

1. It is interesting to see that some monsoon indices show the Indian monsoon is strengthened in the LGM, while other indices show the opposite. Since the paper is run with prescribed SST/sea ice, do similar results appear in the coupled run of iCESM?

Thank you for the suggestion. Following this, we calculated the monsoon indices from the coupled iCESM model results for the LGM and PI climates (Tierney et al., 2020; Zhu and Poulsen, 2021). The results are largely consistent between the coupled model and the fixed-SST iCESM simulations. The monsoon indices calculated from the coupled simulations also indicate a strengthened Somali Jet, enhanced barotropic shear, strengthened Mascarene High, reduced Vertically Integrated Moisture Transport (VIMT), and increased vertical shear of the zonal winds in the LGM simulation (Fig.1 in this document). The ΔTT, however, shows an opposite response compared to the iCESM results, although the magnitude of the response is small (–2% LGM-PI in fixed SST iCESM versus +5% in the coupled simulation).

The following sentences will be added in Section 3.3.1 after the discussion on monsoon indices "*We also calculated corresponding monsoon indices from the coupled iCESM*

*model results for the LGM and PI simulations (Fig. S9: Tierney et al., 2020; Zhu and Poulsen, 2021). The coupled model results, except for ΔTT index, are largely consistent with the fixed-SST iCESM simulations, with the indices indicating a strengthened Somali Jet, enhanced barotropic shear, an intensified Mascarene High, reduced VIMT, and enhanced vertical shear of the zonal winds in the LGM simulation. The ΔTT shows an opposite response compared to the fixed SST iCESM results, although the magnitude of the response is small in both (-2% in iCESM versus +5% in the coupled simulation).* "

Figure 1 is shown at the end of this document and will be included as a supplementary figure in the revised document.

2. It is also interesting to see that the changes in Pacific moisture contribution are very important in regulating the positive precip d18O anomaly in India. Why is that? Is it because less moisture comes from the Pacific due to the strengthened westerlies in the subtropics? The authors can add a discussion for this. Also, how to understand that precipitation is less in India, but the condensation effect is negative in Figure 8c?

   a) **Pacific Moisture contribution**

   The reduced moisture contribution from Pacific is a key control on the positive $\delta^{18}O_{precip}$ anomaly over India because water vapor from Pacific is the most isotopically depleted among dominant source regions. Our decomposition analysis shows that the reduction in the relative contribution of moisture from the North Pacific results in a +0.4‰ enrichment. As the reviewer pointed out, this reduction is consistent with the LGM circulation response, where a relatively warmer western tropical Pacific and a cooler Indian subcontinent strengthen a Walker-like circulation. This enhances convection over the western Pacific. This, together with enhanced westerlies, weakens east-to-west moisture transport into South Asia. Consequently, India receives less vapor from this highly depleted source during the LGM, enriching $\delta^{18}O_{precip}$ even though total monsoon precipitation decreases.

In the revised manuscript, in the Discussions and conclusions section after the sentence "..*the simulated LGM enrichment is due to a reduced relative contribution from distant, isotopically depleted moisture sources and decreased rainout from Indian Ocean sources.*", we will include : "*A key component of this enrichment is the reduction in Pacific moisture contribution in the LGM, which is highly isotopically depleted. Our decomposition analysis shows that the reduction in the relative contribution from the North Pacific alone contributes approximately +0.4‰ enrichment (Fig. 8d). This reduced Pacific moisture contribution also reflects the circulation response in the LGM. The enhanced Walker-like circulation, together with the enhanced westerlies intensify convection over the western Pacific and weaken moisture transport from the Northern Pacific into South Asia.*"

b) **Figure 8c and the condensation term:** In our decomposition, the condensation term reflects the local isotopic enrichment in precipitation relative to ambient vapor, not the change in precipitation amount. Our results suggest that the local isotopic enrichment in precipitation during condensation in the LGM is slightly weaker than in the PI, leading to a net negative value in Fig. 8c. This is likely associated with generally reduced convection and condensation over the sink region in the LGM. However, this aspect warrants further investigation in future studies. Our results show that the amount effect does not control the LGM $\delta^{18}O$ enrichment, which instead arises mainly from changes in moisture source contributions and reduced rainouts.

The manuscript will be modified to incorporate this explanation as "*The condensation term in our framework does not reflect changes in precipitation amount, but rather the enrichment associated with condensation. Therefore, a negative value for the effect suggests a weaker enrichment during condensation in the LGM, likely related to generally reduced humidity, convection, and condensation. Furthermore, cold conditions can reduce re-evaporation of precipitation (Worden et al. 2007), which usually leads to a more isotope-enriched precipitation. However, these aspects warrant further investigation in future studies.*"

3. After the authors draw conclusions that Indian precip d18O in the LGM cannot be explained by the amount effect but by moisture source changes, how should we interpret speleothem/marine sediment d18O changes in the Indian monsoon region in the LGM?

In our simulations, the LGM enrichment of Indian monsoon $\delta^{18}O_{precip}$ is not controlled by the intensity of local precipitation, but primarily by changes in moisture source contributions and weaker rainout along trajectories from the Indian Ocean and reduced relative contribution from distant, isotopically depleted sources (Pacific, Atlantic, South China Sea). Hence, we suggest LGM $\delta^{18}O$ changes recorded in speleothems and marine sediments from the Indian monsoon region should be interpreted mainly as reflecting changes in large-scale circulation and moisture source/pathway, rather than as a direct proxy for intensity of local precipitation. We suggest a more positive LGM $\delta^{18}O$ value in these archives can coexist with weaker monsoon precipitation and is best viewed as an integrated signal of both atmospheric circulation and source-driven changes in the hydrological cycle.

We will rephrase the discussion on the paleo-proxy interpretations as *"Our results imply that $\delta^{18}O_{precip}$ changes simulated in the LGM are driven primarily by changes in moisture source contributions and weaker rainouts, and large-scale circulation changes. Hence, we suggest $\delta^{18}O$ in paleoclimate archives from the Indian monsoon region is better considered as an integrated representation of changes in the hydrological cycle, rather than a direct measure of local rainfall amount. This also shows that combining proxy archives with isotope-enabled climate model simulations is crucial for accurately interpreting past monsoon changes."*

Equation (1): Some letters in the subscripts of P are not subscribed.

Thank you for pointing this out. We have corrected the subscript formatting in Equation (1).

Line 251: P J et al. 2020: Please write the complete last name

Corrected to "Vidya et al. 2020" as suggested.

Equation (3): Please write the equation in the equation mode.

Thank you for noting this. We have re-written Equation (3) in equation mode.

Lines 327-337: Though the averaged precip d18O has a large bias, it seems that precip d18O in most places of India is close to the observations.

Thank you for this observation. We will emphasise this in the revised manuscript as *"It should be noted, however, that while the simulated domain-mean $\delta^{18}O_{precip}$ shows a negative bias, the spatial pattern compares well with several GNIP stations across India."*

Figure 2c: "Source contribution" is kind of misleading here. I thought it was the contribution due to moisture source changes (like you defined in Line 293). Please consider changing it to other words.

Thank you for the suggestion. We agree that the term "source contribution" in Figure 2c could be misleading, as in the decomposition framework, this term refers to the contribution due to moisture source changes. In Figure 2c, we show the isotopic signatures of each moisture source at the sink ($\delta^{18}O_{pi}$), and not the source-change contribution term. Hence, we have modified the title and Y-axis labels for Fig. 2b and 2c for clarity. We have updated these labels as follows:

2b Title: Contribution to precipitation by moisture sources (JJAS)

2b Y-axis: Fraction of total precipitation (%)

2c Title: $\delta^{18}O$ of source-tagged precipitation (JJAS)

2c Y-axis: $\delta^{18}O_p$ [tag] (‰)

Lines 439-442: What is the region of the "weakening of westerley"? It is not clear to me. The strengthening of the westerly is obvious.

Thank you for pointing this out. Although the strengthening of the westerly is the dominant pattern in the LGM relative to the PI, a weakening of the westerly winds is simulated over the Northern Arabian Sea, as seen mainly in Fig. S7b. This weakening is limited to a small region in the northern Arabian Sea, whereas strengthening dominates across most of the monsoon domain. To make this clear, we will restructure the first two

paragraphs of Section 3.3.1 as below, so that the regional strengthening is discussed first, followed by a clarification that the weakening occurs only locally over the Northern Arabian Sea. Underlined are the new additional sentences on monsoon indices from coupled iCESM results, following the previous suggestion.

*"A regional intensification of the low-level westerly winds is simulated across the central and southern parts of India and the Bay of Bengal (Fig. 3, Fig. S7b). This regional intensification of the monsoon circulation is captured by several monsoon circulation indices used in this study- an increased vertical shear of zonal winds, strengthening of the Somali Jet, and enhanced barotropic shear (Fig. S9b, d, f, respectively). We suggest the enhanced westerly circulation in parts of the monsoon region, especially the Somali jet, is influenced by a stronger Mascarene high in the Southern Indian Ocean (Fig. S9e, S8b) that enhances the pressure gradient between the Indian land and the Southern Indian Ocean by ~2 mb (Fig. S8b, S9e). The strengthened Mascarene high is likely associated with the sea ice extension and cooling in the Southern Indian Ocean during the LGM (Fig. S1b, c). This is in agreement with the positive relationship between the ISM circulation and pressure gradient between the Indian monsoon region and the Mascarene high, suggested by several previous studies (Kripalani et al. 2007; Vidya et al. 2020; Azhar et al. 2023). However, the tropospheric temperature gradient (ΔTT), shows a weakening by 2.5% in the LGM. This indicates a weaker thermal forcing of the monsoon, likely due to enhanced cooling in the northern box used for the estimation of ΔTT (Fig. S2b, S5a), in the LGM simulation. We also calculated corresponding monsoon indices from the coupled iCESM model results for the LGM and PI simulations (Fig. S9: Tierney et al., 2021; Zhu and Poulsen, 2020). The coupled model results, except for ΔTT index, are largely consistent with our fixed-SST iCESM simulations, with the indices indicating a strengthened Somali jet, enhanced barotropic shear, an intensified Mascarene High, reduced VIMT, and enhanced vertical shear of the zonal winds in the LGM simulation. The ΔTT shows an opposite response compared to the iCESM results, although the magnitude of the response in both is small (-2% in iCESM versus +5% in the coupled simulation).*

*Although this strengthening dominates over most of the monsoon domain, a weakening of the low-level (850 hPa) westerly circulation and wind speeds towards land is simulated over the Northern Arabian Sea (Fig. 3, Fig. S7b). This weakening is driven by*

*substantially weakened land-ocean thermal contrast (Fig. S5a, larger cooling over the land) and pressure gradients (S8b; Roxy et al. 2015; Weldeab et al. 2022). Surface cooling over the Indian subcontinent (domain mean -4.5°C; Table S2) in the LGM is approximately 1°C greater than the sea surface temperature cooling in the neighbouring northern parts of Arabian Sea, which is consistent with the lower heat capacity of land, leading to more pronounced cooling and enhanced surface pressure over land."*

Line 467: missing right parentheses

Thank you, it's fixed in the manuscript.

Line 485: It seems that the vertical advection term can also explain some drying of western India.

Thank you for the observation. We have now incorporated this point in the manuscript:

*"The analysis (Fig. 4) shows that the drying over most parts of the ISM domain is primarily driven by the reduction in horizontal moisture advection (Fig. 4b), reflecting both reduced atmospheric humidity and weakened moisture transport. Further, the vertical advection term (Fig. 4c) also contributes to drying over north and west regions in India, indicating suppressed upward motion in this region."*

Line 623: Speleothem -> speleothem
Thank you for noting this. Corrected.

**References:**
Tierney, J. E., Zhu, J., King, J., Malevich, S. B., Hakim, G. J., & Poulsen, C. J. (2020). Glacial cooling and climate sensitivity revisited. Nature, 584(7822), 569–573. https://doi.org/10.1038/s41586-020-2617-x

Worden, J., Noone, D., Bowman, K. et al. Importance of rain evaporation and continental convection in the tropical water cycle. Nature 445, 528–532 (2007). https://doi.org/10.1038/nature05508

Zhu, J., & Poulsen, C. J. (2021). LGM climate forcing and ocean dynamical feedback and their implications for estimating climate sensitivity. Climate of the Past, 2021, 253-267. https://doi.org/10.5194/cp-17-253-2021

[Figure]

*Fig.1 Monsoon circulation Indices calculated from the monthly means of PI and LGM climates from coupled iCESM simulations (Tierney et al., 2020; Zhu and Poulsen,*

*2021). The data used are long-term monthly means from the last 100-years of these simulations. The geographical areas for the calculations are shown in Figure S2b. JJAS mean value of differences between LGM and PI as (LGM-PI) in % is shown in the right top of each panel.*

*__Panel a)__ shows the tropospheric temperature gradient (ΔTT) between the northern box (10°N - 35°N, 30°-110°E) and the southern box (15°S - 10°N, 30°-110°E). __Panel b)__ shows the vertical shear of zonal winds (u in m/s) calculated as the change between U200 and U850 (U200-U850) averaged over the region (10°N-30°N, 50°E-95°E). __Panel c)__ shows the hydrological index, calculated by averaging the Vertically Integrated Moisture Transport (VIMT) in the Indian Ocean-Arabian Sea region, [20°S-30°N, 40°E-100°E. __Panel d)__ shows Somali jet speed index, calculated as the square root of twice the area-averaged kinetic energy of 850 hPa horizontal winds over the region (5°S-20°N, 50°E-70°E). __Panel e)__ shows the mean sea-level pressure difference between the Mascarene high (MH; 20°S-40°S, 45°E-100°E) and the wider Indian summer monsoon region (10°N-35°N, 45°E-100°E. __Panel f)__ shows the barotropic shear estimated over 10°N-26°N, 70°E-90°E.*